# *Cis*-regulatory evolution of Wnt family genes contributes to a morphological difference between silkworm species

Kenta Tomihara[1][◕], Ana Pinharanda[2,3][◕], Young Mi Kwon[2], Andrew M. Taverner[4], Laura S. Kors[5], Matthew L. Aardema[4,6], Julia C. Holder[2], Lin Poyraz[2], Takashi Kiuchi[1‡]* Peter Andolfatto[2‡]*

**1** Department of Agricultural and Environmental Biology, Graduate School of Agricultural and Life Sciences, The University of Tokyo, Tokyo, Japan, **2** Department of Biological Sciences, Columbia University, New York, New York, United States of America, **3** Department of Immunology and Infectious Diseases, Harvard T.H. Chan School of Public Health, Boston, Massachusetts, United States of America, **4** Lewis-Sigler Institute for Integrative Genomics, Princeton University, Princeton, New Jersey, United States of America, **5** Department of Biology, Barnard College, Columbia University, New York, New York, United States of America, **6** Department of Biology, Montclair State University, Montclair, New Jersey, United States of America

◕ These authors contributed equally to this work.
‡ These authors co-supervised this work.
* pa2543@columbia.edu (PA); kiuchi@g.ecc.u-tokyo.ac.jp (TK)

## Abstract

Closely related species often exhibit distinct morphologies that can contribute to species-specific adaptations and reproductive isolation. One example is Lepidopteran caterpillar appendages, such as the "caudal horn" of Bombycoidea moths, which have evolved substantial morphological diversity among species in this group. Using interspecific crosses, we identify the genetic basis of the caudal horn size difference between *Bombyx mori* and its closest relative *Bombyx mandarina*. The three largest of eight QTL account for one third the mean horn length difference between the species. The largest of these, on chromosome 4, encompasses a conserved Wnt family gene cluster, key upstream regulators that are well-known for their roles in morphological diversification in animals. Using allele-specific expression analysis and CRISPR/Cas9 knockouts, we show that tissue-specific *cis*-regulatory changes to *Wnt1* and *Wnt6* contribute to the species difference in caudal horn size. This kind of modularity enables highly pleiotropic genes, including key upstream growth regulators, to contribute to the evolution of morphological traits without causing widespread deleterious effects.

## Introduction

Understanding the genetic mechanisms underlying morphological evolution has long been a central goal in evolutionary biology [1–3]. This pursuit has yielded a growing

**Data availability statement:** Code and analysis scripts as well as Supplemental Data tables are available at https://doi.org/10.5281/zeno-do.17833042. Whole-genome sequencing data available in the NCBI Sequence Read Archive under BioProject accession PRJNA1380446. Allelic-specific expression amplicons and corresponding metadata can be found in https://doi.org/10.5281/zenodo.17833042.

**Funding:** This work was supported by National Institutes of Health grants R01 GM112758, R01 GM114093, and R01 GM115523 from the National Institute of General Medical Sciences to PA, and by JSPS KAKENHI grant numbers JP20J22954 to KT and JP20H02997 to TK. The funders had no role in study design, data collection and analysis, decision to publish, or preparation of the manuscript. PA and AMT received salary support from NIH grants R01 GM112758, R01 GM114093, and R01 GM115523. AP received salary support from NIH grants R01 GM112758 and R01 GM115523. YMK received salary support from NIH grant R01 GM115523. KT received salary support from JSPS KAKENHI grant JP20J22954.

**Competing interests:** The authors have declared that no competing interests exist.

**Abbreviations:** ASE, allele-specific expression; BIC, Bayesian information criterion; crRNA, CRISPR-RNA; gDNA, genomic DNA; IBB, inverted beta-binomial; QTL, quantitative trait locus; RT-qPCR, reverse transcription quantitative PCR.

number of cases in which the genetic basis of morphological traits that differentiate populations and species has been identified [4]. However, it is increasingly clear that the genetic architecture of morphological evolution is both complex and varied. In some cases, dramatic phenotypic changes can be attributed to a few loci of large effect; in others, they involve many genes of smaller effect. Studies of the genetic basis of pigmentation trait variation between insect species, for example, often reveal a relatively simple oligogenic basis [5–7]. By contrast, the genetic foundations of more complex morphological, physiological, and behavioral traits remain less well resolved. A related debate concerns whether evolutionary changes in these complex traits are primarily driven by modifications in high-level regulatory genes or in their downstream targets, and whether regulatory or protein-coding sequence changes play the dominant role. Addressing these questions is key to building a comprehensive picture of the genetic basis of morphological evolution [8–10].

Insects have evolved a wide range of morphological adaptations, including projections used for combat [11], mate choice [12], digging [13], sensing [14], and defense [15]. Among these, lepidopteran caterpillars exhibit an impressive range of protruding structures, such as setae, spines and horns. The evolution of these structures across different species has resulted in remarkably diverse caterpillar morphologies (Fig 1; S1 Table). One notable example is the "caudal horn," a conspicuous fleshy projection found at the posterior end of Bombycoidea caterpillars (Fig 1B; S1 Table). The caudal horn exhibits substantial variation in size, shape, and color among different species in the family, suggesting potential adaptive importance of this structure. Little is known about its function, but it has been speculated that in some species it may play a role in crypsis or in sensory capacity [16]. Specifically, some Bombycidae have been reported to move their caudal horn when disturbed [17]. While for other Lepidoptera, caterpillars with horn-like projections rapidly beat them when approached by parasitoids, suggesting a potential role for such horns in defense [18].

The mechanisms underlying evolutionary gains and losses of morphological innovations, like the caudal horn, are often inaccessible to genetic analysis due to reproductive barriers between species exhibiting different traits. In this study, we take advantage of the fact that the domesticated silkworm *Bombyx mori* and its closest relative *Bombyx mandarina* (both members of the Bombycoidea group, Fig 1) produce fertile hybrids. *B. mori* is believed to have been domesticated ~4–10 thousand years ago [19–21] from a *B. mandarina*-like ancestor that may have started diverging from extant *B. mandarina* lineages more than 1 million generations ago [22]. During this time, *B. mori* and *B. mandarina* have diverged in a variety of morphological, physiological, and behavioral traits. Several of these traits, including size, loss of flight, silk yield, and larval foraging behavior are likely to be directly associated with the domestication [23–25]. However, for other traits—such as larval caudal horn length, olfactory capacity, and pigmentation—their adaptive significance remains unclear, although in the case of pigmentation, darker coloration in *B. mandarina* may contribute to crypsis, while its reduction in *B. mori* might facilitate detection by farmers ([26–28]; Figs 1B and S1; S1 Table).

Given the broad diversity in larval appendages among Lepidoptera, of which diversification of the caudal horn among the Bombicoidea is a part, we use interspecific

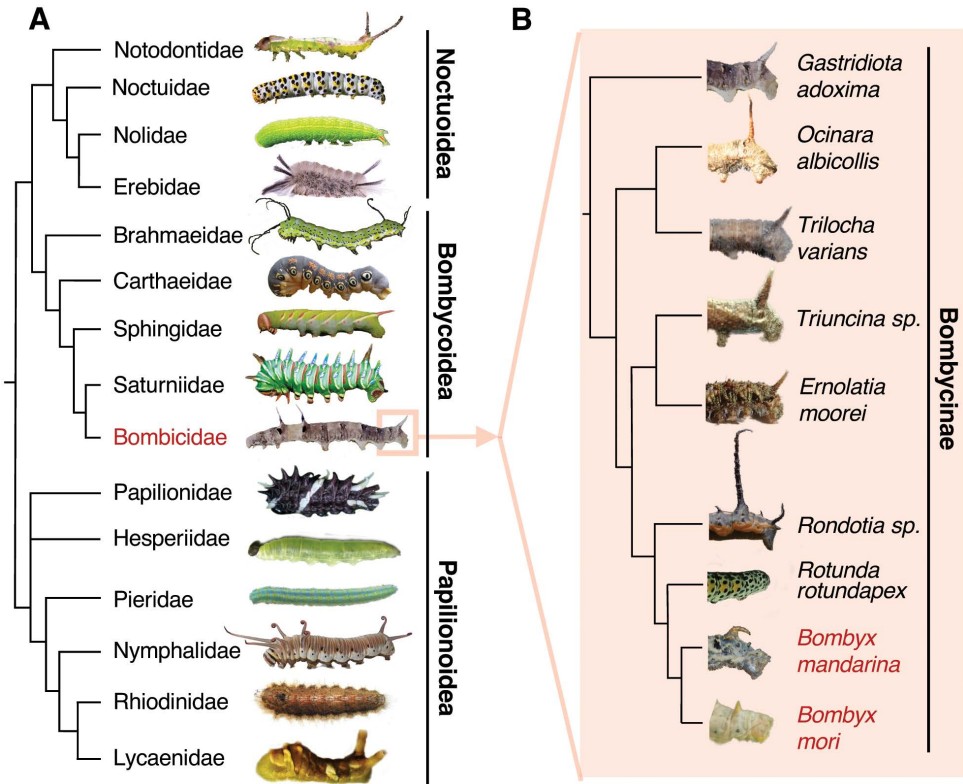

**Fig 1. Evolutionary diversification of Lepidopteran larval appendages. (A)** Representative morphological diversity among Lepidopteran caterpillars. Phylogenetic relationships are indicated by the cladogram and family names for each representative species are indicated. *Bombyx* belongs to the Bombicidae (highlighted in red). **(B)** Representative caudal horn diversity among species in the Bombycinae sub-family. Shown are the posterior larval segments for each species; the cladogram depicts their phylogenetic relationships [58–64,88]. A full list of photo credits is provided in S1 Table. Data can be found in DOI https://doi.org/10.5281/zenodo.17833042.

crosses and advances in high-throughput genotyping to investigate the genetic basis of the caudal horn length difference between *B. mori* and *B. mandarina*. Further, using gene expression studies and targeted knock-outs, we identify causative loci underlying the largest QTL contributing to the caudal horn length difference between species. Our work sheds light on how novel species-specific morphological traits can evolve over relatively short evolutionary timescales.

## Results

### Multiple QTL underlie caudal horn length divergence

The caudal horn structure on the eighth segment of *Bombyx mori* caterpillars is substantially smaller than that of its wild sister species *B. mandarina* (Figs 1B and 2A inset). To quantify this difference, we measured caudal horn lengths of fifth instar larvae using a landmark-based approach (Materials and methods, S1A Fig). The caudal horn of *B. mandarina* Sakado strain ($3.51 \pm 0.08$ mm) is over 3-fold longer than that of *B. mori* p50T strain ($0.94 \pm 0.11$ mm) (Fig 2A). This length difference is consistent across seven additional *B. mori* strains (S1B Fig). The mean caudal horn length for p50T/Sakado F1 hybrids is intermediate ($2.28 \pm 0.09$ mm) suggesting that some alleles contributing to longer caudal horn length in *B. mandarina* are partially dominant. Within species, body size and caudal horn size exhibit positive allometry, as often observed for morphological traits. In contrast, the opposite pattern is observed between species, with *B. mori* caterpillars having a shorter horn than *B. mandarina*, despite the former being substantially larger [25] (S1D Fig).

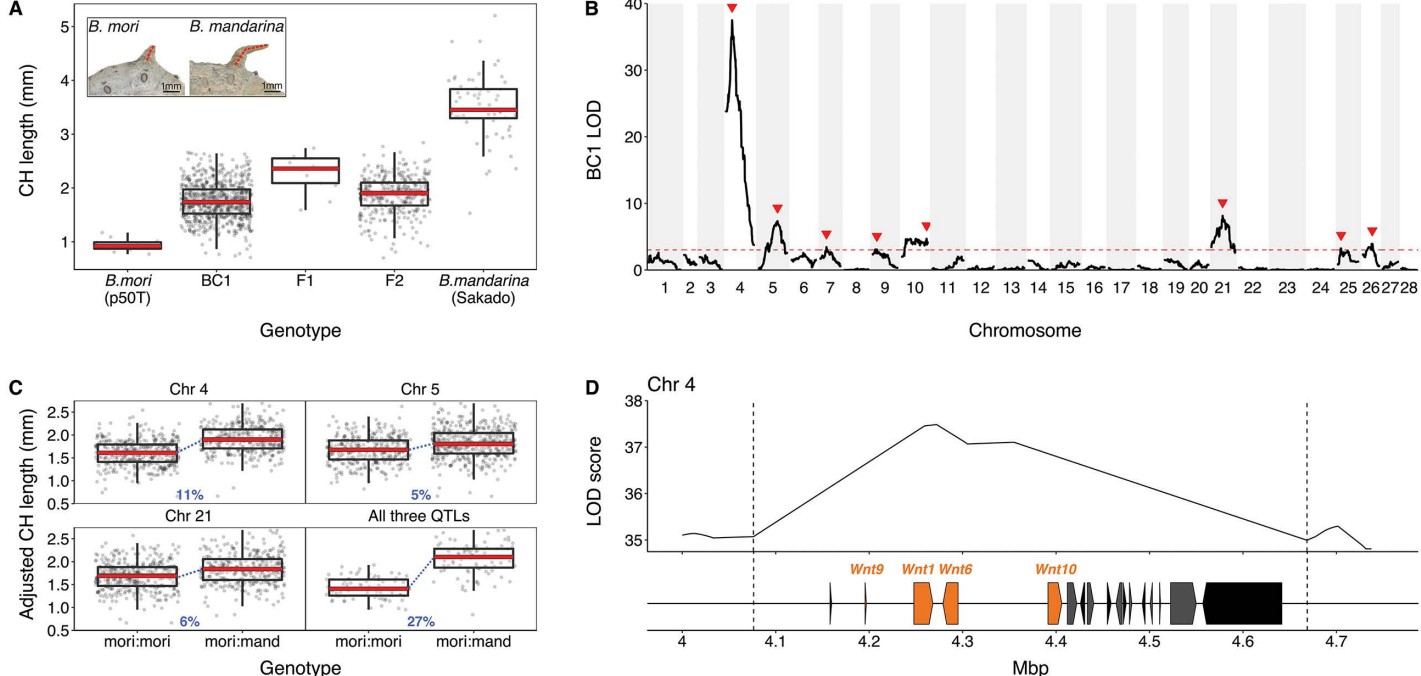

**Fig 2. QTL analyses for caudal horn length. (A)** Distributions of caudal horn (CH) lengths in *Bombyx mori* p50T and *Bombyx mandarina* (Sakado) parental species and hybrids. F1 hybrids are the progeny of p50T females × Sakado males; BC1 are the progeny of p50T females and F1 males; F2 are the progeny of F1 hybrids generated by crossing *B. mori* p50T males and *B. mandarina* Sakado females. The inset shows representative parental phenotypes (see also S1 Fig). Red dashed lines represent the length measurements for these individuals. Photographs by Kenta Tomihara. **(B)** QTL LOD profile plots for caudal horn length based on the BC1 mapping panel. LOD is shown in black with a red dashed line indicating the estimated significance threshold (3.02) determined by 1,000 phenotype permutations. Red arrows indicate significant QTL peaks. **(C)** Caudal horn length distributions of BC1 individuals based on marker genotypes immediately under the three highest LOD peaks on chromosomes 4, 5, and 21. Caudal horn lengths are adjusted for larval weight and sex. The last panel shows caudal horn distributions of BC1 individuals whose genotypes are jointly homozygous *B. mori* or jointly heterozygous at all three QTL. Effect sizes are indicated in blue and represent the change in length as a proportion of the mean length difference between parental species. **(D)** Annotation of the chromosome 4 QTL interval. Boxes indicate annotated genes within the 1.5 LOD interval (delimited by dashed lines) estimated in the BC1 QTL analysis (S4 Table). Wnt family genes are colored in orange. Coordinates are in Megabase pairs (Mbp). Data can be found in https://doi.org/10.5281/zenodo.17833042.

To investigate the genetic basis of caudal horn length divergence between species, we generated two hybrid mapping populations. For the first, *B. mori* females were crossed with *B. mandarina* males to generate F1 hybrids, and F1 males were then backcrossed to *B. mori* females, producing a first-generation backcross (BC1) panel of 694 individuals. For the second mapping population, *B. mori* males were crossed with *B. mandarina* females to generate F1 hybrids, and F1 males and females were intercrossed to produce an F2 panel of 327 individuals (Materials and methods, S2 Fig). The phenotype distribution of BC1 progeny continuously spans the range of phenotypes between *B. mori* p50T and p50T/ Sakado F1 hybrids (Fig 2A), as expected for trait variation that is encoded by several causative loci.

Quantitative trait locus (QTL) analyses of these two mapping populations reveal multiple loci contributing to caudal horn length divergence between species. Analysis of BC1 mapping population identifies eight QTL with significant effects on caudal horn length (Fig 2B). The largest effect QTL is localized to a 600-kilobase interval on chromosome 4 (Chr4). This QTL accounts for 18% of the variance among BC1 progeny and 11% of the mean length difference between species (S2 Table; Fig 2C). The three largest effect BC1 QTL (Chr4, Chr5, and Chr21) jointly account for 27% of the mean length difference between species. The remaining QTL have smaller, marginally significant effects and are less well spatially resolved (Fig 2C; S2 Table). Caterpillar weight (a proxy for size) and sex also contribute significantly to the variance in

caudal horn length among BC1 progeny. Interestingly, all eight QTL have effects in the same direction, *i.e.,* the *B. mandarina* allele is associated with longer mean caudal horn length (QTL Sign Test *p*-value = 0.008; Figs 2C and S3). The biased direction of QTL effects implies that difference in caudal horn size between species is not likely to be solely attributable to genetic drift [29].

Since F1 male individuals were backcrossed only to *B. mori* females (S2 Fig), analysis of the BC1 panel can only detect loci for which the *B. mandarina* allele has a substantial effect in heterozygotes. To alleviate this limitation, we performed QTL analysis of the F2 mapping population. This F2 QTL scan found only one locus with significant effects on caudal horn length, Chr4 (S3 Table; S3 Fig), which overlaps the largest QTL observed in the BC1 analysis and shows consistent effect size with the BC1 Chr4 QTL. The effect size of the F2 Chr4 QTL is consistent with the findings from the BC1 cross. Furthermore, we independently verified the Chr4 QTL by phenotyping consomic (and semi-consomic) introgression strains of Chr4 (see S4 Fig). Conspicuously absent from the F2 QTL scan are the additional QTL detected in the BC1 mapping panel. The smaller mapping panel for the F2 analysis may explain the relatively coarse spatial resolution of the Chr4 QTL and the absence of the seven other QTL detected in the BC1 analysis. However, the proportion of *B. mandarina* ancestry on BC1 QTL-bearing chromosomes—excluding Chr4—is a significant predictor of caudal horn length (S3 Table, *p*-value = 3.4e−7), suggesting that loci on these chromosomes contribute to the trait difference despite not reaching significance in the F2 QTL analysis.

## Evidence for additional loci beyond mapped QTL

The above analyses provide strong evidence for at least eight loci contributing to the caudal horn length difference between species. However, how many more loci might contribute remains unclear. Notably, the proportion of *B. mandarina* ancestry on the remaining non-QTL-bearing chromosomes is also a significant predictor of caudal horn length in the F2 mapping panel (S3 Table, *p*-value = 4.3e−10), consistent with a model in which many small-effect loci also contribute to the morphological difference between species. To investigate this further, we carried out a regression analysis of caudal horn length as a function of per chromosome *B. mandarina* ancestry (S5 Fig). This analysis reveals that the proportions of *B. mandarina* ancestry on Chr3 and Chr6 are significant predictors of increased caudal horn length in the BC1 mapping panel, in addition to the QTL already identified by QTL mapping (S5B Fig). Even after accounting for QTL-bearing chromosomes, a significant positive correlation between caudal horn length and *B. mandarina* ancestry persists in the F2 mapping population. This indicates that additional small-effect caudal horn-shortening alleles on non-QTL-bearing chromosomes have accumulated in the *B. mori* lineage. We estimate that *B. mandarina* ancestry on non-QTL bearing chromosomes accounts for 8.5% of the variance among F2 caudal horn lengths (S3 Table). Thus, while the horn trait difference is clearly dominated by an oligogenic architecture, an unknown number of additional loci on non-QTL-bearing chromosomes also make a substantial contribution to between-species divergence in this trait..

## Two Wnt family genes exhibit caudal horn-specific *cis*-regulatory evolution

The 1.5 LOD (Logarithm of the Odds) interval for the BC1 Chr4 QTL region contains 17 annotated genes (S4 Table). Among these is a cluster of four Wnt family genes (Fig 2D), a family of secreted growth factor proteins with diverse roles in morphological development in animals [30,31]. *Wnt* genes are highly conserved across animals, and this cluster (*Wnt9*, *Wnt1*, *Wnt6,* and *Wnt10*) is syntenic with orthologous genes in the *Drosophila melanogaster* genome (S4 Table). The QTL region is syntenic in the two *Bombyx* species, with no evidence of an inversion or large insertions/deletions.

Given the presence of the *Wnt* gene cluster in the Chr4 QTL region, we speculated that differences in tissue-specific regulation of one or more of these *Wnt* genes may account for species differences in caudal horn size. To test this hypothesis, we investigated patterns of tissue-specific expression for 13 of the 17 genes in the Chr4 QTL 1.5 LOD interval, excluding 3 of retroviral origin and 1 with no diagnostic marker (S4 Table). First, we carried out RNA-seq to identify segment-specific differential expression between the two species (S5 Table). We examined early-stage caterpillars

(second instar, L2) because the difference in caudal horn length between the species is already visible when they hatch (S1C Fig), becoming more pronounced in later larval stages (S1A and S1D Fig). We detected 1,306 transcripts with segment-specific expression divergence between species (S2 Data). Out of these, 121 transcripts overlap the identified QTL regions. Interestingly, this analysis reveals that only *Wnt6* exhibits significant species- and tissue-specific differential expression, with a 2-fold upregulation in the eighth abdominal (A8) segment of *B. mandarina* relative to *B. mori* in the Chr4 QTL (S6A Fig; S5 Table).

However, bulk-tissue RNA-seq-based differential expression analysis has limited statistical power for genes with low expression levels like this Chr4 cluster of Wnt family genes (S6A Fig). Moreover, between-species comparisons of gene expression can be difficult to interpret due to species-specific differences in the *trans*-regulatory environment and other differences in development (e.g., cell number, the timing of development, etc.). To identify genes with *cis*-regulatory divergence between species, we examined allele-specific expression (ASE) in F1 hybrids, which compares the expression of two alleles in a common *trans*-environment [32].

To assay ASE, we used a targeted PCR-based approach (see Materials and methods). We further dissected the A8 segment into the caudal horn structure itself (CH) versus the remaining "non-horn" A8 tissue, with the goal of improving tissue resolution beyond the level of segment (Fig 3A). This analysis reveals that only two genes, *Wnt1* and *Wnt6*, exhibit caudal horn-specific ASE in the fifth instar (L5) of F1 hybrid caterpillars (Fig 3B; S6 Table). Specifically, both genes exhibit a caudal horn-specific down-regulation of the *B. mori* allele relative to the *B. mandarina* allele. A similar trend is observed in the F1 hybrid caterpillars at the first instar (L1) stage when the between-species caudal horn length difference is less pronounced (S6B Fig; S6 Table). Genomic DNA (gDNA) samples from F1 hybrids were processed in the same way as cDNA, and in all cases the *B. mandarina:B.mori* allele ratios did not differ significantly from 1, confirming the absence of amplification bias (see Materials and methods for details, S3 Data). These results show that the *cis*-regulatory elements of *Wnt1* and *Wnt6* have diverged between these species, suggesting that differences in their tissue-specific expression may contribute to the species difference in caudal horn length. Notably, the Wnt1 and Wnt6 proteins of *B. mori* and *B. mandarina* are identical in amino acid sequence, indicating that any functional differences between the species involving these genes are solely due to *cis*-regulatory differences in gene expression (S4 Table).

To examine temporal patterns of *Wnt1* and *Wnt6* expression, we used reverse transcription quantitative PCR (RT-qPCR) to measure relative transcript abundance in integumentary tissue from caudal-horn-bearing and control segments (A8 and A7, respectively) of *B. mori* and *B. mandarina*. This analysis (Figs 3C and S6C) revealed several species- and tissue-level differences in expression dynamics. First, both *Wnt1* and *Wnt6* are generally expressed at higher levels in the caudal horn bearing A8 segment. Second, expression of both genes is consistently higher in *B. mandarina* than in *B. mori*, especially in the A8 segment. Third, this species difference emerges earlier for *Wnt6* and peaks for both genes at the onset of the molt preceding the L5. These findings suggest that ASE differences measured during the fifth instar (Fig 3B) may underestimate the divergence present at the end of the fourth instar. Specifically, *Wnt1* expression was highest in the caudal segments of *B. mandarina* on day 3 of the fourth instar, and at this time point *Wnt1* levels were also elevated in the caudal segments of both species relative to non-caudal segments. Together, the results support the idea that reduced *Wnt1* and/or *Wnt6* expression in the caudal segment of *B. mori* may contribute to its shorter horn phenotype.

At the L5 stage, species-level expression differences (Fig 3C) closely match ASE differences measured in the caudal horn of F1 hybrids (Fig 3B). Notably, the differential expression *Wnt1* exhibits 2.3-fold higher expression in *B. mandarina* relative to *B. mori*, compared to a 2.2-fold *B. mandarina*-allele bias in F1 hybrids. Similarly, *Wnt6* exhibits 3.1-fold higher expression in *B. mandarina* relative *to B. mori* and a 3.0-fold *B. mandarina*-allele bias in F1 hybrids. This concordance indicates that *cis*-regulatory divergence is sufficient to explain most of the observed species-level expression differences for both genes, with little evidence for an important *trans*-regulatory contribution.

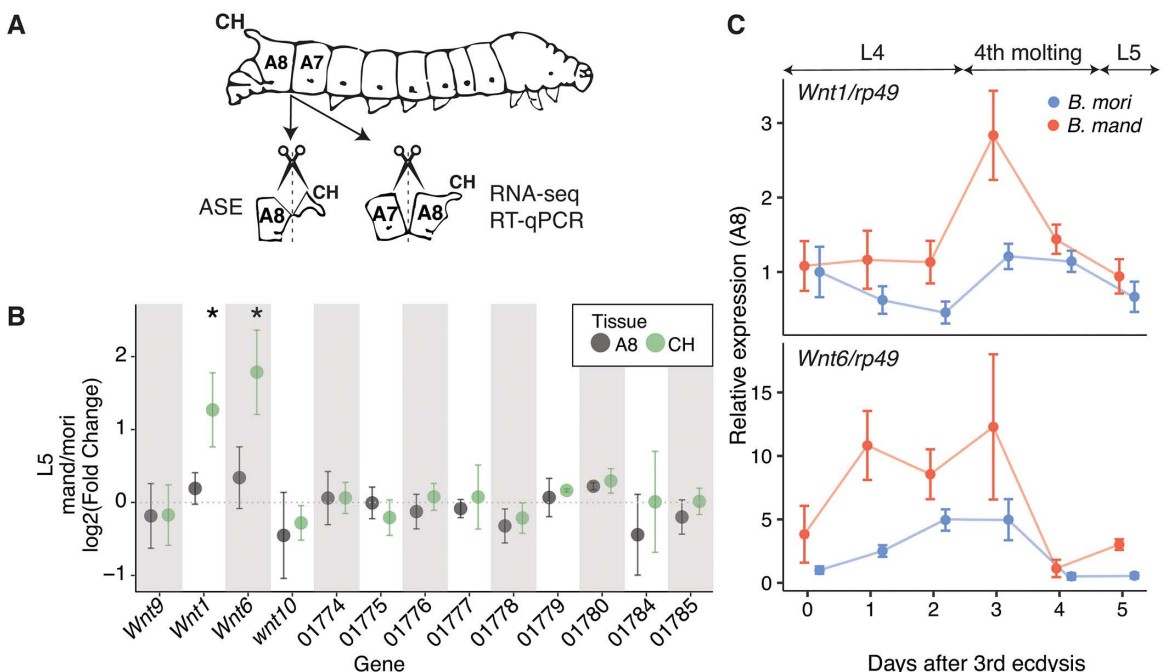

**Fig 3.** *Cis*-regulatory divergence underlies species- and tissue-specific *Wnt1* and *Wnt6* expression patterns during larval development. **(A)** Schematic of the sampling scheme for gene expression analyses. Total RNA was extracted from caudal horn (CH, segment A8) and non-horn tissue of the eighth abdominal segment (A8) for RNA-seq (S6A Fig) and targeted ASE (Figs 3B and S6B). Total RNA from the integument of segments A7 and A8 was extracted for RNA-seq (S6A Fig) and the temporal expression analysis (Figs 3C and S6C). Diagram based on Takeda (2009) [89]. **(B)** Allele-specific expression patterns in the caudal horn of *Bombyx mori* (p50T) × *Bombyx mandarina* (Sakado) F1 hybrids of L5 (final) instar caterpillars. Plotted for each gene in the QTL interval (in the same order, left to right), is the log2(fold-difference) in the average *B. mori* and *B. mandarina* allele counts for the caudal horn portion of segment A8. Values larger than 0 indicate higher expression of the *B. mandarina* allele and whiskers represent two standard errors (corresponding to 95% confidence intervals among biological replicates). "CH" shows the caudal horn tissue and "A8" the non-horn tissue portion (Fig 3A). A corresponding analysis of L1 (first) instar caterpillars is shown in S6B Fig. Significance of ASE differences between the caudal and non-caudal portions of segment A8 were evaluated using the inverted beta-binomial test (IBB) test for paired count data and significant differences are indicated with asterisks (Bonferroni corrected, S6 Table). **(C)** Temporally-elevated expression of *Wnt1* and *Wnt6* in segment A8 of *B. mandarina*. Temporal expression of *Wnt1* and *Wnt6* in the integument tissue of segment A8 during the transition between fourth and fifth instar stages of *B. mori* and *B. mandarina* caterpillar development. A corresponding analysis of segment A7 is shown in S6C Fig. Relative *Wnt1* and *Wnt6* expression levels (setting *B. mori* A8 at day 0 of 4th instar = 1) were normalized to those of *ribosomal protein 49* (*rp49*). Whiskers represent two standard errors among replicates. Data can be found in https://doi.org/10.5281/zenodo.17833042.

## Functional analyses reveal that *Wnt1* and *Wnt6* contribute to caudal horn length evolution

While we have established that both *Wnt1* and *Wnt6* are down-regulated in the caudal horn of *B. mori* relative to *B. mandarina*, expression studies alone cannot determine whether one or both genes have a causal effect on horn growth. Thus, to functionally validate the role of *Wnt1* and *Wnt6* in the evolved caudal horn difference between species, we used allele-specific CRISPR/Cas9-mediated gene knockouts of *B. mandarina* alleles [26]. We crossed injected (G0) females of semi-consomic line T04 (chr4^mand/chr4^mori) to *B. mori* p50T males to generate G1 larvae. G1 females whose *B. mandarina*-derived *Wnt1* or *Wnt6* is disrupted by small frameshift causing premature stop codon were crossed to p50T males to establish *B. mandarina*-allele-specific knockout mutants (S7 Fig). Knocking out the *B. mandarina*-derived allele of *Wnt1* in hybrids results in a 51% reduction in caudal horn length (Tukey's HSD, *p*<0.001; Fig 4A). Knocking out the *B. mandarina*-derived allele of *Wnt6* also results in a reduction in horn length, but the effect (a 7% reduction) is substantially smaller (Tukey's HSD, *p*<0.05; Fig 4A). Notably, with the exception of caudal horn length, we did not notice any pigmentation or morphological changes in larvae with *Wnt1* and *Wnt6* knock-outs. If *B. mori* represents the derived state, our results suggest that the

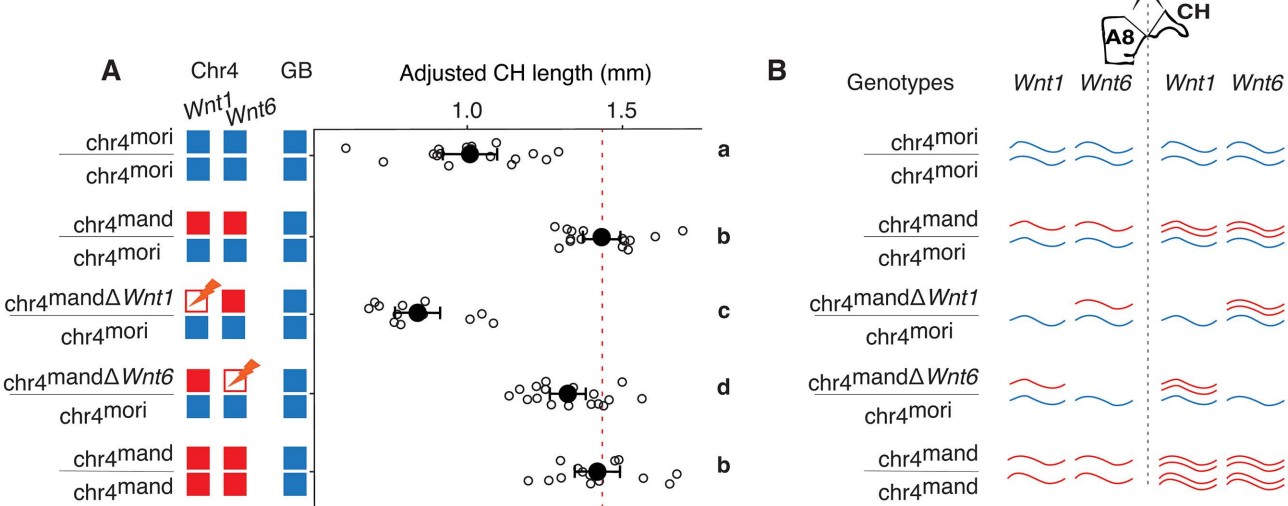

**Fig 4. *Wnt1* and *Wnt6* CRISPR/Cas9-mediated *Bombyx mandarina*-allele-specific knockouts significantly reduce caudal horn length. (A)** Distributions of caudal horn lengths for larvae at day 0 of the fifth instar (L5). Diagrams to the left indicate the genotypes of the strains being compared (Chr4 = chromosome 4; GB = genomic background). Blue corresponds to the p50T *Bombyx mori* background; and red to the *B. mandarina* Sakado background. chr4$^{mandΔWnt1}$/chr4$^{mori}$ and chr4$^{mandΔWnt6}$/chr4$^{mori}$ correspond to CRISPR/Cas9-mediated *B. mandarina*-allele-specific knockouts of *Wnt1* and *Wnt6*, respectively. White points indicate individual caterpillar horn lengths, corrected for caterpillar weight (a proxy for size). Black points and whiskers correspond to means and standard errors. Letters on the right indicate groups that differ significantly in caudal horn length based on a one-way ANOVA followed by Tukey's HSD test (family-wise confidence level = 95%). A significant positive allometric relationship was observed between body weight and caudal horn length (ANCOVA: effect of weight, $p = 2.9 × 10^{-16}$), with a significantly steeper slope in *B. mandarina* than in *B. mori* (interaction term, $p = 0.016$), see S1 Fig. **(B)** Conceptual representation of relative *Wnt1* and *Wnt6* transcriptional abundances for each genotype. Expression results (Fig 3) indicate that *Wnt1* and *Wnt6* have elevated transcript abundance in caudal horn tissue of stage L5 *B. mandarina* caterpillars. Knock-outs of the *B. mandarina* allele are expected to decrease (but not eliminate) expression of *Wnt1* or *Wnt6* relative to all other genotypes in the entire animal. Data can be found in DOI https://doi.org/10.5281/zenodo.17833042.

down-regulation of *Wnt1* in *B. mori* contributes substantially to its reduced caudal horn length relative to *B. mandarina* (Fig 4B).

## Discussion

### The role of Wnt1 and Wnt6 in morphological divergence between species

We investigated the molecular mechanisms underlying the diversification of a conspicuous larval appendage structure, the caudal horn, in Bombycoidea caterpillars. Our findings highlight how morphological divergence between species can result from tissue-specific regulatory changes affecting major upstream developmental genes. Specifically, we show that *cis*-regulatory downregulation of two Wnt family genes, *Wnt1* and *Wnt6*, in the developing caudal horn of *B. mori* is the single largest contributing factor to the reduction in caudal horn length in this species. Although our analyses focus on later larval stages, reduced horn length in *B. mori* relative to *B. mandarina* is already evident at the neonate stage (S1 Fig), suggesting that the causal regulatory differences arise earlier in development, potentially during embryogenesis.

Wnt family genes are well known for their roles in limb and appendage development [30,33,34] and, more broadly, Wnt signaling is associated with cell proliferation and tissue growth across many developmental contexts [35,36]. Despite this, there are few clear examples in which evolutionary changes to Wnt genes themselves drive morphological evolution. Prior work has shown that *cis*-regulatory evolution of *Wnt1* and *WntA* can underlie pigmentation differences among insect species [37–39]. Beyond pigmentation differences, recent work in *Heliconius* butterflies [40] and *B. mori* [41] showed that Wnt

signaling can modulate appendage extension through regulation of the downstream transcription factor *aristaless*. However, whether these effects are mediated by *cis*-regulatory changes at Wnt family genes remains unclear. To our knowledge, the role of *Wnt1* and *Wnt6* in *Bombyx* caudal horn divergence is the first example in which *cis*-regulatory evolution at Wnt family genes has been directly linked to divergence in a morphological structure (S7 Table).

The regulation of expression and functional roles of Wnt family genes are diverse and complex across animals [42–44]. In insects, *Wnt1* and *Wnt6* share high sequence similarity, are physically adjacent in the genome, and exhibit overlapping expression domains. These features have led to the hypothesis that the two genes are at least partially co-regulated and may act redundantly in some developmental contexts [39,45,46]. Consistent with this view, a shared enhancer for *Wnt1* and *Wnt6* has been identified in *D. melanogaster* [47]. Since both *Wnt1* and *Wnt6* are downregulated in the *B. mori* caudal horn relative to *B. mandarina*, it may be that the causal *cis*-regulatory changes involve a shared horn-specific enhancer. Allele-specific knockouts further reveal that both *Wnt1* and *Wnt6* contribute to horn development—though to differing degrees—supporting the idea of partial redundancy (Fig 4). The larger effect of *Wnt1* on caudal horn size in allele-specific knockout experiments suggests that it is more functionally potent or dosage-sensitive in this context.

Together, these findings add to a small but growing list of examples showing that *cis*-regulatory changes in major upstream developmental genes can underlie morphological differences, beyond pigmentation traits, between closely related species and domesticated animals (S7 Table). Our results also underscore how regulatory modularity enables the resolution of developmental trade-offs associated with morphological trait divergence. The tissue-specific downregulation observed in *B. mori* illustrates how spatially restricted regulatory changes can decouple traits, allowing selection to act on horn size without potentially negative pleiotropic effects. Such modularity is a key feature of evolvability and exemplifies the utility of tissue-specific enhancers in facilitating adaptive change.

## Large effect loci contribute to the evolution of the caudal horn in *Bombyx*

Although domestication differs from natural selection, it nonetheless offers a powerful framework for studying rapid evolutionary change under strong and often directional selective pressures [48]. While some domestication traits are mendelian, others involve more complex, quantitative architectures. Our QTL mapping analysis suggests that, in addition to *Wnt1*/*Wnt6*, at least seven more loci with smaller effects contribute to horn length variation. The three largest QTL jointly account for 27% of the difference in mean caudal horn between species, indicating that multiple loci of moderate-to-large effect explain a substantial portion of the phenotype. The oligogenic architecture of this domestication-associated trait difference resembles that of at least some traits shaped by natural selection, such as pelvic spine reduction in stickleback [49] and sexually dimorphic pigmentation in *Drosophila* [50].

Additionally, we find that QTL loci act in concert with a substantial measurable genetic component residing outside of mapped QTL loci. This finding raises the question of the true degree of polygenicity of the caudal horn trait difference. Seven of the eight detected QTL have small estimated effect sizes, likely reflecting loci with true effect sizes that are even smaller [51,52] and close to the threshold of detectability for this mapping panel. This implies that the true number of contributing loci could be substantially larger. While few studies have quantified the relative contribution of the genomic background in detail, it is increasingly likely that a substantial contribution small effect loci to trait differences is a common, if often overlooked, feature of morphological divergence between species (see, for example, [53]).

## Evolutionary forces shaping caudal horn evolution

The selective pressures driving caudal horn reduction in *B. mori* remain uncertain. Direct selection on horn length during domestication is unlikely. However, the consistent direction of effect across all eight QTL—where *B. mandarina* alleles increase horn length—argues against a purely neutral model [29,54]. One possibility is that horn reduction occurred as a correlated response to selection on other traits. Alternatively, the horn may have become maladaptive in the domesticated

environment. The repeated gain and loss of horn-like structures across wild Bombycoidea species suggests that natural selection has repeatedly acted on this trait, potentially targeting similar developmental pathways.

Changes in Wnt family gene expression have previously been implicated in morphological differences among closely-related species [37–39,55]. In Lepidoptera, *Wnt1* expression has been linked to variation in pigmentation [56] and to the formation of protruding structures [57]. Our results extend these findings by showing that tissue-specific downregulation of *Wnt1* in the developing caudal horn, also a protruding structure in caterpillars, is linked to *cis*-regulatory evolution of *Wnt1* itself, rather than changes to *trans*-acting factors. A key next step will be to identify the precise *cis*-regulatory elements responsible for *Wnt1*/*Wnt6* downregulation in *B. mori*. It is plausible that similar regulatory modules have been repeatedly co-opted for morphological innovation across Lepidoptera (Fig 1). Comparative studies in other species could reveal whether parallel evolution of Lepidopteran morphology has been driven by convergent changes in *Wnt* gene regulation.

## Conclusions

This study integrates quantitative genetics with functional analyses to reveal the molecular basis of a conspicuous morphological trait difference distinguishing closely related species. We demonstrate that spatially precise regulatory changes in key developmental genes—*Wnt1* and *Wnt6*—underlie horn reduction in *B. mori*. These results highlight how *cis*-regulatory evolution can drive morphological divergence while minimizing pleiotropic effects, emphasizing the role of enhancer modularity in enabling adaptive evolution. This principle is likely to be broadly applicable across systems and taxa in evolutionary developmental biology.

## Materials and methods

### Phylogenetic relationships

A cladogram for representative Lepidopteran species was constructed based on phylogenetic reconstructions of the Noctuoidea [58], the Bombicoidea [59], and the Papilonidea [60] and a broader phylogeny including all three groups [61]. A cladogram for the Bombicidae was reconstructed based on several recent phylogenetic reconstructions of members of this group [62–64].

### Sample collection

*Bombyx mori* strains c51, f35, g53, o56, p20, p21, p22, T04, and CT04 were provided from Kyushu University supported by the National BioResource Project (https://shigen.nig.ac.jp/silkwormbase/). *B. mori* strain p50T and *B. mandarina* strain Sakado were maintained at the University of Tokyo. For the BC1 mapping panel, F1 males were generated by crossing *B. mori* (p50T) females to *B. mandarina* (Sakado) males (S2 Fig). We crossed F1 males to females *B. mori* p50T, generating BC1 progeny. For the F2 mapping panel, *B. mori* p50T males were crossed to *B. mandarina* Sakado females to produce an F1 generation, and then F1 males and females were crossed to produce an F2 generation (S2 Fig). We phenotyped and genotyped 696 BC1 and 327 F2 caterpillars as outlined below.

### Rearing and phenotyping

Larvae were reared on white mulberry (*Morus alba*) leaves or an artificial diet SilkMate PS (NOSAN, Japan) at 25 °C with a 12-hour light:12-hour dark:light cycle. We weighed and photographed fifth instar larvae at day 0 for subsequent phenotyping. Caudal horn lengths were measured from photographs by marking two points at the base of the horn and a third point at the tip of the horn using either a custom MATLAB script or ImageJ [65]. The microscope S9D (Leica Microsystems, Germany) equipped with a MC 170 HD camera (Leica Microsystems) was used to take photographs. For each horn, we defined a triangle extending upward from the base of the horn using three points (1, 2, 3), and an additional segment defined by a 4th point at the center of the horn (S1A Fig). Length (*L*) was then calculated as,

$$L = \frac{|(X_2 - X_1)(Y_1 - Y_3) - (X_1 - X_3)(Y_2 - Y_1)|}{\sqrt{(X_2 - X_1)^2 + (Y_2 - Y_1)^2}} + \sqrt{(X_4 - X_3)^2 + (Y_4 - Y_3)^2},$$

where $X_k$ and $Y_k$ are coordinates of point k. The phenotype distributions of the BC1 and F2 are consistent with normal distributions (Shapiro–Wilk test $p = 0.14$ and $p = 0.10$, respectively).

### Genome-wide ancestry assignment

We used Multiplexed Shotgun Genotyping [66] to carry out genome-wide ancestry assignment of recombinant progeny. Genomic DNA was extracted from the legs of the adults using the Quick-DNA extraction kit (Zymo Research, USA). Whole genome short-read Illumina-style libraries were generated using a Tn5-based "tagmentation" method [67]. For each individual, 4–15 ng of genomic DNA (measured using Synergy MX Microplate Reader [BioTek, USA]) was digested with Tn5 that was pre-charged with Illumina-style adapters. Individuals were indexed by PCR amplification (18 cycles) using combinations of Illumina-compatible i5 and i7 indexed PCR primers. Library sequencing for BC1 was performed on HiSeq 2500 (Illumina, USA) at the Princeton University Genomics Core Facility, yielding 75 nucleotide single-end reads. Libraries for the F2 were sequenced on a HiSeq 4000 (Illumina, USA) at the Weill-Cornell Medical College Genomics Resources Core Facility, yielding 150 nucleotide paired-end reads, and read 1 was used for downstream analysis. Reads were de-multiplexed and adapters removed using Trim Galore (bioinformatics.babraham.ac.uk/projects/trim_galore/). The average number of reads per individual was 572,000 and individuals with fewer than 100,000 reads were discarded.

A *B. mori* p50T strain genome assembly (BHWX01000001-BHWX01000696, 2016) from Silkbase [68], and a *B. mandarina* Sakado strain genome assembly (GCA_030267445.2, NCBI) were used as parental reference genomes. Repetitive regions were identified and masked in the *B. mori* and *B. mandarina* genomes using RepeatMasker v4.1.2 with the species set to "bombyx." We also masked sites that were polymorphic in either p50T or Sakado strains. To identify polymorphic sites within each strain, paired-end short-read Illumina reads of a p50T male (DRR064025; NCBI) and two Sakado samples (DRR059710, DRR093001; NCBI) were adapter-trimmed using Trim Galore v0.6.10 and aligned to the *B. mori* genome (bwa v0.7.17) [69] and variants were called using bcftools mpileup v1.16 [70]. Sites were removed based on the following quality filters: MQ < 30, GQ < 25, removing any SNPs within 5 bp of an indel (--SnpGap 5:indel), and where the read-depth was less than half (<0.5×) or more than two times (>2×) the average total read depth.

MSG was run with *B. mori* designated as parent1 and *B. mandarina* as parent2, using the following parameters: deltapar1 = 0.01, deltapar2 = 0.05, recRate = 28, rfac = 0.0001, and priors set to (0.25, 0.5, 0.25) for F2 crosses and (0, 0.5, 0.5) for BC1 crosses. For each individual, genotypes were assigned as missing (*i.e.*, "NA") if the posterior probability was below 90%. Additionally, we masked genomic regions between ancestry switches within an individual that occurred less than 1 Mbp apart and sites with more than 20% missing genotypes across individuals within each cross. This filtering yielded a set of 1,595,348 markers shared between the BC1 and F2 datasets. To reduce redundancy, we applied marker thinning using pull_thin (https://github.com/andrewtaverner/pull_thin) using default settings, except: numMarkers = 5000, ignoreNan = True, and autosome_prior = NA. A final set of 140,000 markers (5000 per chromosome) was retained for QTL analyses.

### Quantitative trait locus (QTL) analysis

We performed two separate QTL analyses using R/qtl [71] with genotype and caudal horn lengths for 327 F2 and 694 BC1 individuals. The Haley-Knott regression method [72] was used for QTL mapping with sex, weight (a proxy for size), and genome-wide *B. mandarina* ancestry as covariates. After an initial QTL detection step excluding ancestry as a co-variate, ancestry was estimated from non-QTL-bearing chromosomes as the proportion of the genome assigned as B. mandarina-derived accounting for diploidy and regions with missing genotypes, Genome-wide 5% significance thresholds were

determined by 1000 phenotype permutations [73]. These were estimated as LOD > 3.03 and LOD > 3.63 for the BC1 and F2 analyses, respectively. For each significant QTL, a 1.5 LOD support interval spanning the maxLOD peak was considered the candidate region [74]. Variance explained was estimated using the function fitqtl [71].

**Differential expression analysis**

Caudal and non-caudal segments (A8 and A7, respectively; Fig 3B) were dissected from caterpillars before the onset of the second (L2) instar. Three replicate pools of eight individuals were generated for each species and segment (except the *B. mori* non-caudal segments, for which there are two replicates). Samples were stored at −80 °C in RNAlater (Thermo Fisher Scientific). RNA was extracted using the RNAeasy kit (QIAGEN) and RNA-seq libraries were prepared using TruSeq RNA Library Preparation Kit v2 (Illumina). Libraries were sequenced on HiSeq 4000 (Illumina) at the Weill-Cornell Medical College Genomics Resources Core Facility, yielding 312 million 150-nucleotide single-end reads. This corresponds to an average of 28 million reads per replicate and tissue. Reads were mapped to the *B. mori* annotated transcriptome [68] using STAR [75]. Read count output from STAR was then imported into DESeq2 [76] for differential expression analysis.

Differential expression analysis was performed to detect transcripts with caudal (A8) and non-caudal (A7) segment-specific expression differences between the species. To this end, the model used to analyze all tissue samples from all replicates was:~Species:Replicate + Species*Tissue, where "Species" is *B. mori* or *B. mandarina* and "Tissue" is caudal (A8) or noncaudal (A7). The first term of the model defines Replicate as a nested variable within Species and accounts for the fact that caudal and non-caudal segments were sampled from the same individuals. Our quantity of interest is the second term, which accounts for potential interactions between Species and Tissue (*i.e.*, species- and tissue-specific differential expression).

**Allele-specific expression (ASE) analysis**

To investigate ASE patterns for the genes within the Chr4 QTL, we extracted RNA from relevant tissues in *B. mori* p50T × *B. mandarina* Sakado F1 hybrids. The caudal segment (A8) was dissected from first (L1) and fifth (L5) larval instars and further dissected into caudal horn (CH) and "non-horn" A8 tissue (Fig 3B). Four replicate pools of 10 individuals were sampled for each segment and instar. Samples were stored in RNAlater Stabilization Solution (Thermo Fisher Scientific), and total RNA was extracted using TRIzol (Thermo Fisher Scientific), RNeasy columns (QIAGEN), and DNaseI (Thermo Fisher Scientific). Total RNA from each sample was used to create cDNA using SuperScript IV VILO (Thermo Fisher Scientific). The conversion to cDNA was repeated three times for each sample, representing technical replicates.

We quantified relative allele abundance using a targeted multiplexed PCR-sequencing approach. For each of 13 genes in the Chr4 QTL interval we tested (see S4 Table), PCR primers were designed to target a genic region that spans at least one SNP distinguishing the two species (S8 Table). We PCR-amplified these amplicons from F1 cDNA samples using the following reaction conditions: 94 °C (30″), 95 °C (15″), 60 °C (30″), 68 °C (45″), 68 °C (5′) for 15 cycles. To allow multiplexed Illumina sequencing, we incorporated a second set of universal i5 and i7 Illumina-style indexes using the following conditions: 72 °C (3′), 94 °C (10″), 62 °C (15″), 68 °C (30″), 68 °C (5′) for 12 cycles. After removing primer dimers with DNA clean & concentrator (D4030), indexed amplicons were pooled and sequenced on a MiSeq (Illumina) using Reagent Micro Kit v2 (Illumina), 300 Cycles (MS-103-1002). Reads were demultiplexed and, for each replicate, we counted the number of reads corresponding to each species-specific SNP.

gDNA was extracted from the same F1 hybrid design as the RNA samples, consisting of four replicate pools of 10 individuals each, using the DNeasy Blood & Tissue Kit (QIAGEN) and assayed using the same primer pairs to test for amplification bias. Allele counts from gDNA did not deviate significantly from a 1:1 *B. mandarina* to *B. mori* ratio (two-sided binomial tests, $P > 0.05$ in all cases), confirming the absence of systematic bias. For cDNA, ASE was quantified as $\log_2$ (*B. mandarina*/*B. mori*) per replicate and means ±2SE were reported for visualization. To formally test whether allele ratios

differed between caudal-horn and non-horn tissues, we used the inverted beta-binomial (IBB) test for paired-count data [73] as implemented in the R countdata package, comparing four caudal cDNA replicates against four non-caudal replicates per gene. *P*-values were corrected using Benjamini–Hochberg procedures. This analysis was used both to confirm whether cDNA allele ratios deviated from the gDNA-derived expectation of 1:1 and to identify significant differences between caudal horn and non-horn tissues. The data relevant to this analysis can be found in https://doi.org/10.5281/zenodo.17833042.

To assess whether genes within the QTL interval differed at the amino-acid level between *B. mori* and *B. mandarina*, we compared protein sequences derived from multiple genome assemblies (*B. mori*: GCF_014905235.1; *B.mandarina*: GCF_003987935.1, GCA_030267445.2). In addition, *B. mandarina* protein predictions were generated by lifting *B. mori* gene annotations onto the *B. mandarina* reference genome (GCA_030267445.2) using liftOver [77] and, where necessary, by reciprocal Exonerate [78] alignments to confirm orthology and exon structure. Protein sequences were aligned using MUSCLE [79], and alignments were inspected in SeaView [80] to identify candidate amino-acid substitutions between species. For each gene, substitutions were evaluated across assemblies. Putative amino-acid differences that were not consistently supported across *B. mandarina* genome assemblies, that had exon annotation differences, or that were absent from short-read sequencing data were considered likely annotation artifacts and excluded from further analysis. Only amino-acid substitutions supported by multiple assemblies were retained.

### Reverse transcription quantitative PCR (RT-qPCR)

To quantify the relative abundance of *Wnt1* and *Wnt6* RNA in a time course of larval development, we used an RT-qPCR approach. Total RNA was isolated from integuments of caudal (A8) and non-caudal (A7) segments of *B. mori* and *B. mandarina* using TRIzol (Thermo Fisher Scientific). Reverse transcription was performed using the oligo (dT) primer and avian myeloblastosis virus Reverse Transcriptase contained in the RNA PCR kit (TaKaRa, Japan). We conducted RT-qPCR using the Kapa SYBR FAST qPCR kit (Kapa Biosystems, USA) and StepOnePlus Real-time PCR System (Applied Biosystems, USA). RNA abundance is quantified relative to *ribosomal protein 49* (*rp49*). Primer sets are listed in S9 Table.

### CRISPR/Cas9-mediated allele-specific knockouts

To evaluate the effects of *Wnt1* and *Wnt6* expression on caudal horn development, we used CRISPR/Cas9-mediated allele-specific knockout approach [26]. Instead of F1 hybrids, we instead used the T04 semi-consomic strain, which is heterozygous *B. mori* (p50T)/*B. mandarina* (Sakado) for Chr4 in an otherwise *B. mori* (p50T) genomic background ([81]; see S4 Fig).

We identified Cas9 PAM-sites that differed in sequence between *B. mori* and *B. mandarina* and designed CRISPR-RNA (crRNA) targeting *B. mandarina*-specific sequences (S6 Fig) [26]. To obtain non-diapausing eggs, we first incubated T04 eggs at 15 °C under continuous darkness until they hatch [82]. The hatched larvae were then reared under continuous 16-h light/8-h dark at 25 °C and crossed adult females to p50T males and collected fertilized eggs. A mixture of crRNA (200 ng/µL; FASMAC, Japan), *trans*-activating crRNA (tracrRNA) (200 ng/µL; FASMAC), and Cas9 Nuclease protein NLS (600 ng/µL; NIPPON GENE) was injected into eggs within 3 hours of oviposition [83]. Injected embryos were then incubated at 25 °C in a humidified Petri dish until hatching. Adult generation 0 (G0) females were crossed with the wild-type p50T males to obtain generation 1 (G1) progeny. Genomic DNA was extracted from the G1 individuals by the HotSHOT method [84], and PCR was performed using KOD One (TOYOBO, Japan) or MightyAmp DNA Polymerase Ver.3 (TaKaRa). Mutations at the targeted site were detected by heteroduplex mobility assay using the MultiNA microchip electrophoresis system (SHIMADZU, Japan) with the DNA-500 reagent kit [85,86]. The PCR products were also cloned into pGEM-T Easy Vector (Promega, USA) and Sanger-sequenced using the FASMAC sequencing service. G1 females carrying a 7-nucleotide deletion in *B. mandarina*-derived *Wnt1* and 2-nucleotide deletion in *B. mandarina*-derived *Wnt6* were crossed to wild-type p50T males, respectively, to establish

*B. mandarina*-allele-specific knockout mutants (S7 Fig). Both mutations result in frameshifts and premature stop codons. Mutant larvae were weighed and the caudal horn length measured as described above, and then genotyped using PCR. RNA oligos and PCR primer sets are listed in S10 Table.

### Effects of chromosome-level ancestry on caudal horn length

We analyzed ancestry and its relationship to caudal horn length using the standardized coefficients from the linear regression model as measures of effect size (R v.4.0.2). Bayesian information criterion (BIC) model selection was used to evaluate the association between ancestry and caudal horn length (S5 Fig) with ancestry on all chromosomes included as independent variables. We adjusted *p*-values using a Bonferroni correction for the 27 chromosomes tested.

### Supporting information

**S1 Fig. Measurement of caudal horn length variation. (A)** Images of L5 larval caudal horns were marked at three locations (colored points) using MATLAB or ImageJ. Two purple points are laid to mark the base of the caudal horn, corresponding to $X_{1,2}, Y_{1,2}$ (see Materials and methods: Rearing and Phenotyping), and the black point marks the vertex $X_3, Y_3$. The height of the triangle is then added to the distance between the black point and the blue point $X_4, Y_4$ located at the tip of the caudal horn. Photographs by Kenta Tomihara. **(B)** Caudal horn (CH) length measurements across various strains of *Bombyx mori* and *Bombyx mandarina* Sakado. The *B. mori* strain used in this study is p50T. Values were not adjusted for weight or sex. **(C)** Neonate (day 0 of first instar) larvae of *B. mori* (left) and *B. mandarina* (right). Red arrows indicate the caudal horns. Scale bars = 1 mm. **(D)** Caudal horn (CH) length and caterpillar weight relationship for *B. mori* (p50T) and *B. mandarina* (Sakado) at L5 stage. Values were not adjusted for weight or sex. Data can be found in [https://doi.org/10.5281/zenodo.17833042](https://doi.org/10.5281/zenodo.17833042).
(EPS)

**S2 Fig. Schematic of the cross design for the BC1 and F2 mapping panels. (A)** The BC1 mapping panel. *Bombyx mandarina* Sakado (Bmand) males were crossed to *Bombyx mori* (Bmori) p50T females to create F1 males. F1 males were then backcrossed to *B. mori* p50T to produce BC1 progeny. Note that the W chromosome of BC1 females (red) is derived from *B. mori* p50T. The large X indicates meiotic crossing over in F1 males. **(B)** The F2 mapping panel. *B. mori* p50T (Bmori) males were crossed to a *B. mandarina* (Bmand) Sakado females to create F1 males. F1 males and females were crossed to produce F2 progeny. Note that the W chromosome of F2 females (black) is derived from *B. mandarina* Sakado. The large X indicates meiotic crossing over in F1 males. Data can be found in [https://doi.org/10.5281/zenodo.17833042](https://doi.org/10.5281/zenodo.17833042).
(EPS)

**S3 Fig. F2 QTL analysis and estimated QTL effect sizes. (A)** Caudal horn (CH) length distributions of BC1 individuals based for markers corresponding to maximum LOD peaks for QTLs on chromosomes 7, 9, 10, 25, and 26. Caudal horn length was adjusted for larval sex and weight. Effect sizes (in blue) are the % mean length difference between heterozygous (mori:mand) and homozygous *Bombyx mori* (mori:mori) genotype, normalized by the mean length difference between the parental species. **(B)** F2 QTL LOD profile for caudal horn length. LOD is shown in blue with the red dashed line indicating the LOD significance threshold of 3.62 determined by 1,000 permutations. **(C)** Superposition of BC1 (black) and F2 (blue) LOD profiles for Chr4. The vertical line and red triangle indicate the max LOD position in BC1. LOD significance threshold for BC1 is 3.02, and F2 is 3.62. **(D)** Caudal horn length distributions of F2 individuals based on marker genotypes immediately under the LOD peak at chromosome 4. Lengths are adjusted for larval weight. Effect sizes are indicated in blue and represent the change in length as a proportion of the mean length difference between parental species. Data can be found in [https://doi.org/10.5281/zenodo.17833042](https://doi.org/10.5281/zenodo.17833042).
(EPS)

**S4 Fig. Analysis of consomic and semi-consomic Chr4 introgression strains. (A)** Schematic diagram showing genome-wide ancestry of semi-consomic (T04) and consomic (CT04) strains. The semiconsomic strain T04 segregates for a *Bombyx mandarina* (Sakado) derived chromosome 4 in an otherwise *Bombyx mori* (p50T) genomic background [81]. In contrast, the consomic CT04 strain is homozygous for the *B. mandarina* (Sakado) derived chromosome 4 in the same *B. mori* (p50T) background. Individual caterpillars were genotyped to identify those with one copy (chr4$^{mand}$/chr4$^{mori}$) versus zero copies (chr4$^{mori}$/chr4$^{mori}$) of *B. mandarina* chromosome 4 (see Materials and methods). The mean caudal horn length of chr4$^{mori}$/chr4$^{mori}$ individuals is 15% shorter than chr4$^{mand}$/chr4$^{mori}$ and chr4$^{mand}$/chr4$^{mand}$ individuals (S4 Fig), which is consistent with the direction, dominance and effect size estimated for the chromosome 4 QTL. **(B)** Caudal horn length distributions among Chr4 genotypes. Caudal horn length was adjusted for larval weight and sex. Effect sizes are indicated in blue and represent the change in length as a proportion of the mean length difference between parental species. A significant difference in corrected caudal horn length was observed between mori:mori and mori:mandarina T04 individuals ($t$ test, $p = 1.0 \times 10^{-8}$; means ± SE: 1.01 ± 0.04 mm versus 1.44 ± 0.03 mm, respectively). Data can be found in https://doi.org/10.5281/zenodo.17833042.
(EPS)

**S5 Fig. Identifying genomic contributions of loci beyond mapped QTL to caudal horn length divergence.** BC1 data shown in panels A, B and C; F2 data shown in panels D, E and F. **(A)** The correlation between caudal horn length and genome-wide ancestry for BC1 (Pearson's $\rho = 0.32$, $p = 1.3e{-}17$). Caudal horn length was adjusted for larval weight and sex. **(B)** The correlation between caudal horn length and genome-wide ancestry excluding chromosomes with significant QTLs for BC1 (Pearson's $\rho = 0.05$, $p = 0.23$). Caudal horn length was adjusted for larval weight and sex. **(C)** The estimated effect sizes of *Bombyx mandarina* ancestry on each BC1 chromosome using a Bayesian information criterion (BIC) model selection approach to select the minimal set of chromosomes that explain caudal horn length. Sex and weight were included as covariates. Bolded chromosomes are chromosomes carrying caudal horn length QTLs. Adjusted p-values are shown for significant chromosomes observed in the model. **(D)** The correlation between caudal horn length and genome-wide ancestry for F2 (Pearson's $\rho = 0.45$, $p = 1.76e{-}17$). Caudal horn length was adjusted for larval weight. **(E)** The correlation between caudal horn length and genome-wide ancestry excluding chromosomes with significant QTLs for F2 (Pearson's $\rho = 0.27$, $p = 6.4e{-}7$). Caudal horn length was adjusted for larval weight. **(F)** The estimated effect sizes of *B. mandarina* ancestry on each F2 chromosome using a BIC model selection approach to select the minimal set of chromosomes that explain caudal horn length. Sex and weight were included as covariates. Bolded chromosomes are chromosomes carrying caudal horn length QTLs. Bonferroni-corrected p-values are shown for significant chromosomes observed in the model. Data can be found in https://doi.org/10.5281/zenodo.17833042.
(EPS)

**S6 Fig. Additional gene expression analyses of parental species and F1 hybrids. (A)** Normalized RNAseq counts from second instar larvae (L2) for non-caudal (A7) and caudal (A8) segments of *Bombyx mori* (Bmori) and *Bombyx mandarina* (Bmand). The asterisk indicates a significant species*segment difference for *Wnt6* (adjusted p-value = 0.023). Numbered genes exclude the prefix "KWMTBOMO", see S4 Table. **(B)** Allele-specific expression of caudal horn (CH) and non-caudal horn (A8) portions of the A8 segment for L1 (first) instar caterpillars (see Fig 3A). The fold-difference between *B. mori* (Bmori) and *B. mandarina* (Bmand) alleles is estimated as log2 of the ratio of the averages among the four replicate pools of 10 individuals. Positive values indicate higher expression of the *B. mandarina* allele. **(C)** Temporal expression of *Wnt1* and *Wnt6* in the A7 ("non-caudal") segment of *B. mori* and *B. mandarina* throughout fourth instar (setting *B. mori* A8 at day 0 of 4th instar = 1). The relative mRNA levels are normalized to that of *ribosomal protein 49* (*rp49*). Error bars represent two standard errors. For caudal segment analysis refer to Fig 3D. Data can be found in https://doi.org/10.5281/zenodo.17833042.
(EPS)

**S7 Fig. Design of CRISPR/Cas9-mediated allele-specific knockouts and breeding and screening strategy.** Gene structures and inset alignments of **(A)** *Wnt1* and **(B)** *Wnt6* with selected crRNA target sites whose PAM sequences are only present in *Bombyx mandarina*. The target sequences are underlined, and the PAM sequences are shown in bold letters. *B. mandarina* specific SNPs are highlighted with green shadings. Deletions introduced by CRISPR/Cas9 system are indicated by dashes. **(C)** Breeding and screening strategy used to generate and phenotype *Wnt* knockouts. T04 individuals (G0) were injected with Cas9-crRNA-tracrRNA complex targeting *Wnt1* or *Wnt6* and crossed to *Bombyx mori* (p50T). G1 progeny were genotyped by heteroduplex mobility assay and Sanger sequenced to identify frameshift mutations. Selected G1 mutants were crossed to *B. mori* to produce G2 larvae. G2s were phenotyped for body weight and caudal horn length, then genotyped using PCR. Because chromosomes other than Chr4 were randomly inherited, off-target effects were assumed to have minimal phenotypic impact. Notably, G2 individuals homozygous for the *B. mori* allele at Chr4 (chr4^mori/chr4^mori) did not display aberrant phenotypes. **(D)** Chr4 genotype, genetic background (GB), and representative phenotype images for G2 individuals. Diagrams to the left indicate the genotypes of the strains being compared (Chr4 = chromosome 4; GB = genomic background). Blue corresponds to the p50T *B. mori* background; and red to the *B. mandarina* Sakado background. chr4$^{mand\Delta Wnt1}$/chr4$^{mori}$ and chr4$^{mand\Delta Wnt6}$/chr4$^{mori}$ correspond to CRISPR/Cas9-mediated *B. mandarina*-allele-specific knockouts of *Wnt1* and *Wnt6*, respectively. Representative lateral views of the caudal horn are shown for each class (scale bar, 1 mm). Photographs by Kenta Tomihara. Data can be found in https://doi.org/10.5281/zenodo.17833042. (EPS)

**S1 Table. Licensing and attribution details for photographs used in Fig 1.**
(XLSX)

**S2 Table. Summary of BC1 QTL analysis.**
(XLSX)

**S3 Table. Summary of F2 QTL analysis.**
(XLSX)

**S4 Table. Annotated genes within the BC1 chromosome 4 QTL interval.**
(XLSX)

**S5 Table. Differential expression analysis between *Bombyx mori* and *Bombyx mandarina* in caudal and non-caudal segments of second instar larvae for genes in the Chr4 QTL interval.**
(XLSX)

**S6 Table. Comparison of allele-specific expression in caudal horn (CH) and non-horn A8 tissue.**
(XLSX)

**S7 Table. Examples of major upstream regulator genes with *cis*-regulatory changes that underlie trait differences between closely related or domesticated animal species.**
(XLSX)

**S8 Table. Primer pairs used for Allele Specific Expression (ASE) analysis in F1 hybrids.**
(XLSX)

**S9 Table. Primers used for RT-qPCR assays.**
(XLSX)

**S10 Table. RNA oligo and PCR primer list used in CRISPR/Cas9-mediated allele-specific knockouts.**
(XLSX)

**S1 Data. Distribution of caudal horn length in *Bombyx mori*, *Bombyx mandarina*, and hybrids ([Fig 2A](Fig 2A)).**
(XLSX)

**S2 Data. Tissue-specific expression divergence between *Bombyx mori* and *Bombyx mandarina* in 2nd instar (L2) larvae.**
(XLSX)

**S3 Data. Raw read counts for *Bombyx mori* and *Bombyx mandarina* alleles without allowing for mismatches.**
(XLSX)

**S4 Data. Mean values plotted in figures for easy cross referencing.**
(XLSX)

**S5 Data. Metadata for ASE experiment ([Fig 3B](Fig 3B)).**
(XLSX)

## Acknowledgments

M. Gutin laid the preliminary groundwork for the project as part of her Ph.D. dissertation [87]. We thank L. Ruiz and N. Sparman for help with insect husbandry and M. Rebeiz, T. Shimada, and S. Shigenobu for helpful comments during the course of the project. We thank Patrick F. Reilly for early involvement in the project, valuable scientific discussions, and feedback on the manuscript. We also thank the Institute for Sustainable Agro-ecosystem Services, The University of Tokyo, for facilitating the mulberry cultivation and the Biotron Facility at the University of Tokyo for rearing the silkworms.

## Author contributions

**Conceptualization:** Kenta Tomihara, Ana Pinharanda, Takashi Kiuchi, Peter Andolfatto.

**Data curation:** Kenta Tomihara, Laura S. Kors, Matthew L. Aardema, Julia C. Holder, Lin Poyraz, Ana Pinharanda, Andrew M. Taverner, Peter Andolfatto.

**Formal analysis:** Kenta Tomihara, Ana Pinharanda, Andrew M. Taverner, Peter Andolfatto.

**Funding acquisition:** Kenta Tomihara, Takashi Kiuchi, Peter Andolfatto.

**Investigation:** Ana Pinharanda, Kenta Tomihara, Andrew M. Taverner, Peter Andolfatto.

**Methodology:** Kenta Tomihara, Ana Pinharanda, Takashi Kiuchi, Peter Andolfatto.

**Project administration:** Peter Andolfatto, Takashi Kiuchi.

**Resources:** Peter Andolfatto, Takashi Kiuchi.

**Supervision:** Takashi Kiuchi, Peter Andolfatto.

**Validation:** Kenta Tomihara, Young Mi Kwon, Peter Andolfatto, Ana Pinharanda.

**Visualization:** Kenta Tomihara, Young Mi Kwon, Ana Pinharanda, Peter Andolfatto.

**Writing – original draft:** Kenta Tomihara, Ana Pinharanda, Takashi Kiuchi, Peter Andolfatto.

**Writing – review & editing:** Kenta Tomihara, Ana Pinharanda, Takashi Kiuchi, Peter Andolfatto.

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
