## [Editor Report · Decision Letter 0]

16 Oct 2025

Dear Dr Andolfatto,

Thank you for submitting your manuscript entitled "Cis-regulatory evolution of Wnt -family genes contributes to a morphological difference between silkworm species" for consideration as a Research Article by PLOS Biology.

Your manuscript has now been evaluated by the PLOS Biology editorial staff [as well as by an academic editor with relevant expertise] and I am writing to let you know that we would like to send your submission out for external peer review.

Once your full submission is complete, your paper will undergo a series of checks in preparation for peer review. After your manuscript has passed the checks it will be sent out for review. To provide the metadata for your submission, please Login to Editorial Manager (https://www.editorialmanager.com/pbiology) within two working days, i.e. by Oct 18 2025 11:59PM.

Kind regards,

Ankiit Ahluwalia,

PLOS Biology

aahluwalia@plos.org

---

## [Decision Letter · Decision Letter 1]

3 Dec 2025

Dear Dr Andolfatto,

Thank you for your patience while your manuscript "Cis-regulatory evolution of Wnt-family genes contributes to a morphological difference between silkworm species" was peer-reviewed at PLOS Biology. It has now been evaluated by the PLOS Biology editors, an Academic Editor with relevant expertise, and by several independent reviewers.

Based on the reviews, we are likely to accept this manuscript for publication, provided you satisfactorily address the remaining points raised by the reviewers. Please also make sure to address the following data and other policy-related requests.

IMPORTANT - please attend to the following:

a) Please address the requests from the reviewers. You'll see that reviewer #1 is already satisfied, but reviewer #2 has a number of requests that should be relatively straightforward. Reviewer #3 is more negative about the overall advance, but after discussion with the Academic Editor, we think that the findings are still appropriate for a PLOS Biology Short Report. The AE simply asks that you ensure that your cis regulatory claims are commensurate with the data.

b) Please provide the accession numbers for the deposited sequencing data.

c) Many thanks for providing the underlying data and code in Github. However, because Github depositions can be readily changed or deleted, please make a permanent DOI’d copy (e.g. in Zenodo) and provide this URL (see below).

d) Please cite the location of the data clearly in all relevant main and supplementary Figure legends, e.g. “The data underlying this Figure can be found in doi.org/10.xxx/zenodo.xxxxx" (I know this looks repetitive, but it makes the Figs and their legends more standalone).

We expect to receive your revised manuscript within two weeks.

*Published Peer Review History*

*Press*

Sincerely,

Ankiit

Ankiit Ahluwalia,

aahluwalia@plos.org

PLOS Biology

Reviewer remarks:

Reviewer #1: This is an elegant, swiftly written article on the quantitative genetics of an original insect trait. The authors used hybrids between domestic and wild strains of silkworms to shed light on the genetic architecture of caudal horns, a caterpillar feature that appears convergent across Lepidoptera lineages.

The QTL analysis is impeccably made and using large broods. It is always interesting to see how a new trait behaves in crosses (F1, F2, backcrosses in either direction).

The large-effect main locus encodes a cluster of Wnt genes, which are iconic developmental genes. The follow-up experiments use qPCR refinement and most importantly, allele-specific CRISPR experiments that are particularly convincing. Wnt1 in particular, and also Wnt6, are clearly contributing to the length of the caudal horns. Supposedly, the Wnt pathway probably drives other cases of cuticule protuberances in lepidopteran caterpillars.

The figures shine by the quality of their quantitative data but could also benefit from pictures to help the reader connect the metrics to morphology, maybe.

Overall I find this article to be a neat example of phenotype-to-genotype analysis with an exciting example supporting a developmental genetic toolkit locus in morphological variation. Thanks to its relative simplicity, it's a good introduction to quantitative genetics and should be an interesting read to students interested in insect evo-devo.

I did not find areas of concern, all in all this is an already quite polished manuscript and I believe it is of general interest.

Reviewer #2: Review of the manuscript: Cis-regulatory evolution of Wnt-family genes contributes

to a morphological difference between silkworm species

Overall thoughts and background:

In this manuscript, the authors aim to elucidate the genetic basis of caudal horn length in Bombycoidea moths. The authors describe how such work can provide insights into the genetic control/evolution of complex morphological traits, phenotypic divergence between closely related species, adaptation, and even reproductive isolation. More specifically, the authors use interspecific crosses between Bombyx mori and its closest relative, B. mandarina, which exhibit different caudal horn lengths, to map the genetic differences underlying these length differences. The authors were able to first identify eight QTL from which the three largest explain around one third of the mean horn length differences between species. The largest of these QTLs houses a Wnt-family cluster, which points the authors to the hypothesis that one or several of these Wnt genes might be controlling the differences in horn length. The authors were able to identify Wnt1 and Wnt6 as candidates due to their differential expression between the caudal horn and abdominal segment 8 in hybrids (B. mandarina allele expressed higher). This was validated across development via qPCR where Wnt1 and Wnt6 are shown to be expressed higher in B. mandarina within the abdominal segment 8 and mostly across the time leading to molting. The authors functionally validated the role of Wn1 and Wnt6 in caudal horn length via CRISPR/Cas9-mediated knockouts. They were able to observe that knockouts of the B. mandarina alleles for both Wnt genes reduced horn length (Wnt1 knockout having a stronger effect), showing their involvement in proper horn length.

In general, this manuscript is well put together, organized, and cohesive. The authors do a great job in guiding the reader and also following the logical steps associated with the specific questions. It provides very detailed and, to a certain degree, mechanistic information on the genetic control of the formation of complex structures (like the caudal horn) in a non-model system, which is very impressive. In summary, the authors showcase the data well, making clear both the findings and relevance. However, I want to point out a handful of major concerns with the presentation and analyses of some of the key datapoints. I outline some of those major concerns below with details on some minor concerns as well. I believe that a revised version of the manuscript tackling some of the general requests should merit publication.

Major Concerns or Adjustments:

1. The first major concern is related to the Wnt1 and Wnt6 CRISPR/Cas9-mediated B. mandarina-allele-specific knockouts- I want to highlight that this is an incredible experiment done by the authors and that can't be underscored. However, to my knowledge, the authors did not provide or show the images from which the quantifications were done. Given that the caudal horn is a morphologically complex structure, this quantification needs images highlighting the resulting phenotype or at least representative data used for the quantification. This is key for the unbiased analyses and to satisfy the curiosity of any reader of the manuscript. If images were recorded to perform the quantification, this should be an easy request to incorporate and will heavily enrich the overall impact of the experiments and manuscript. Such images could even be nested within the figure where the quantifications are done, in case the authors want to avoid making a new image.

2. Did the authors ever notice an effect on other morphological metrics when knocking out Wn1 and Wnt6? Again, as the author mentioned, this is a fleshy/complex structure, and I know the starting data details for the mapping are accounting variation based on length, but given their effect on that metric, I was wondering if the authors looked at any other metrics. Like measuring the circularity of the base of the caudal horn, or if there are any pigmentation changes (given the Wnt family's known history of co-regulating shape and pigmentation). Adding details about this would be informative, especially from the angle of learning about the genetic underpinnings of building complex morphological structures, which would line up with what the authors refer to in the introduction. This is also a relevant point for the discussion.

The authors point out a handful of times how we know a lot from Wnt genes controlling morphological divergence between species, mostly from examples in the pigmentation world. There are several scenarios in which such details also interconnect with appendage growth. I feel it is a missed opportunity that the authors did not report on pigmentation (at least for the CRISPR experiment), given that there are small pigmentation differences between Bombyx mori and B. mandarina based at least on the pictures shown in Figure 1. I also understand this request might come off as seeking out new analyses. I want to emphasize that showing images, as mentioned above, would already satisfy this by just adding a couple more lines into the results/discussion and providing the representative data to enhance the relevance of the findings more widely. Again, if not, other differences are noted, stating that those other metrics were lacking differences would be very informative.

Minor Concerns or Adjustments:

1. Line- 126 is one of the first places to include the word backcross at least for one of the first BC1 or a bit more details to ease with readability for this section.

2. Similarly to the major concern 1. I consider that more transparency on the way this structure (caudal horn) is being measured for the main text would be of benefit. Currently, Figure S1 is the only place where we see those relevant details, which are the starting base for all the work that follows. I would include that image or a version of it earlier in the main text before the difference is mapped, or bring it up in the same text location, but not as a supplemental file.

3. The text of the genotypes in Figure 4 is way too small. It was quite difficult to read them, even when zooming in.

4. Statement of line 235 is very interesting and relevant. Was this from data not shown by the authors, or is it lacking a reference?

5. Line 278: The statement can be more specific, highlighting time as well, given that the authors have more details about when the dynamics are changing.

6. Just want to bring to the author's attention that in Heliconius butterflies, it has recently been suggested that Wnt signaling can modulate both appendage extension and pigmentation via regulation of the gene aristaless (Bayala et al. 2023). I think this is a reference point that enhances several of the mechanistic claims presented by the authors and ties really well, given what we know about aristaless from Bombyx within its involvement in antennal branching (Ando et al, 2018).

7. Line 283- Would be good to break the statement between flies and butterflies.

8. Increased Wnt activity is associated with proliferation and cell growth in most documented scenarios, which may be a point to include in the discussion.

9. Line 351- Other species within Moths, or more widely?

10. Line 391- Details of the imaging setup? Are they present somewhere else?

11. Figures 2 and 4 would benefit from a bigger font.

Reviewer #3: The authors present a study of underlying genetic causes for the difference in length of a posterior appendage, the "caudal horn" of the caterpillar of silk worms in the genus Bombyx. They create hybrids of the domesticated silk worm Bombyx mori and a relative, B. mandarina, and use QTL analysis to identify loci influencing species-specific differences in caudal horn length (B. mandarina's is about 3x longer than B. mori's). Major effects are mapped to putative cis-regulatory regions of the Wnt1 and Wnt6 genes, with many minor-effect loci also implicated. Functional testing using CRISPR-mediated mutation of these two genes supports the idea that reduced levels of the two Wnts in B. mori contributes to its shorter caudal horn phenotype.

From a technical standpoint, the study is well executed, and the data are convincing. It's a great illustration of how to find genes involved in morphological divergence. However, without a more in-depth molecular analysis of actual cis-regulatory elements mediating Wnt levels, it is not particularly enlightening in terms of biological mechanism. It is quite well accepted that regulatory differences can be responsible for phenotypic divergence—the authors even include a Table with other examples. While there continues to be debate over the extent to which regulatory vs. coding differences have the largest impact, the basic role of regulatory changes is not disputed. There are even other examples of Wnt involvement. Therefore, while these data may be exciting to those particularly interested in lepidopteran morphology or developmental roles of Wnt genes, I don't find this study to be of high interest generally.

According to the journal, "PLOS Biology Short Reports are defined by novelty and interest of the phenomena being studied." Both of these criteria are quite modest for this study.

---

## [Editor Report · Decision Letter 2]

6 Jan 2026

Dear Dr Andolfatto,

Thank you for the submission of your revised Short Report "Cis-regulatory evolution of Wnt family genes contributes to a morphological difference between silkworm species" for publication in PLOS Biology. On behalf of my colleagues and the Academic Editor, Abderrahman Khila, I am pleased to say that we can in principle accept your manuscript for publication, provided you address any remaining formatting and reporting issues. These will be detailed in an email you should receive within 2-3 business days from our colleagues in the journal operations team; no action is required from you until then. Please note that we will not be able to formally accept your manuscript and schedule it for publication until you have completed any requested changes.

PRESS

Sincerely,

Ankiit Ahluwalia,

PLOS Biology

aahluwalia@plos.org